# An underground drip water monitoring network to characterize rainfall recharge of groundwater at different geologies, environments, and climates across Australia

Andy Baker[1], Margaret Shanafield[2], Wendy Timms[3], Martin S. Andersen[4], Stacey Priestley[5] and Marilu Melo Zurita[6]

[1]School of Biological, Earth and Environmental Sciences, UNSW Sydney, NSW 2052, Australia
[2]College of Science and Engineering, Flinders University, GPO Box 2100 Adelaide, Australia
[3]School of Engineering, Deakin University, Waurn Ponds, Victoria 3216, Australia
[4]School of Civil and Environmental Engineering, UNSW Sydney, NSW 2052, Australia
[5]Drought Resilience Mission, CSIRO, Adelaide, Australia
[6]School of Humanities and Language, UNSW Sydney, NSW 2052, Australia

*Correspondence to*: Andy Baker (a.baker@unsw.edu.au)

**Abstract.** Understanding when and why groundwater recharge occurs is of fundamental importance for the sustainable use of this essential freshwater resource for humans and ecosystems. However, accurately capturing this component of the water balance is widely acknowledged to be a major challenge. Direct physical measurement identifying when groundwater recharge is occurring is possible by utilizing a sensor network of hydrological loggers deployed in underground spaces located in the vadose zone. Through measurements of water percolating into these spaces from above, we can record the potential groundwater recharge process in action. By using automated sensors, it is possible to precisely determine when recharge occurs (which event, month, or season, and for which climate condition). Combined with daily rainfall data, it is possible to quantify the 'rainfall recharge threshold', the amount of rainfall needed to generate groundwater recharge, and its temporal and spatial variability. Australia's National Groundwater Recharge Observing System (NGROS) provides the first dedicated sensor network for observing groundwater recharge at an event-scale across a wide range of geologies, environments, and climate types representing a wide range of Australian hydroclimates. Utilizing tunnels, mines, caves, and other subsurface spaces located in the vadose zone, the sensors effectively record 'deep drainage', water that can move beyond the shallow subsurface and root zone to generate groundwater recharge. The NGROS has the temporal resolution to capture individual recharge events, with multiple sensors deployed at each site to constrain the heterogeneity of recharge between different flow paths, and to quantify (including uncertainty bounds) rainfall recharge thresholds. Established in 2022, the network is described here together with examples of data being generated.

## 1 Introduction

For the sustainable use of our essential groundwater resources an understanding of when and why groundwater recharge occurs is of fundamental importance (Gleeson et al., 2012). At the catchment or regional scale, estimates of the diffuse rainfall recharge of groundwater used by the international research community are generally inferred from water chemistry and water level information that we obtain from boreholes (e.g., Crosbie et al., 2010; Jasechko and Taylor, 2015). However, even with the latest methodologies and syntheses (e.g., Jasechko, 2019; Crosbie and Rachakonda, 2021; Berthelin et al., 2023), these recharge estimates are typically uncertain long-term averages (e.g., annual averages). Such methods, and data, cannot identify specific recharge events and are often

affected by groundwater abstraction. In contrast, techniques that measure infiltration at the surface can do so at
the event scale but provide limited understanding about the portion of water that may recharge the groundwater, as much of this infiltrated water will be removed by evapotranspiration (Shanafield and Cook, 2014). Geophysical estimates of total subsurface water can be obtained from the Gravity Recovery and Climate Experiment and Follow On products (GRACE/GRACE–FO), with recent attempts to use this data to derive large-scale quantified groundwater recharge estimates (Hu et al., 2023). This approach is computationally intensive, has a coarse spatial
resolution (~200 x 200 km) and is not able to separate the recharge process from soil moisture storage. The large uncertainty means that the international research community uses the word 'estimate' when referring to recharge calculations using existing methods (Healy, 2010). Crucially, existing methods are 'snapshots in time' that are not long enough or of sufficient temporal resolution to identify the relationships between climate and groundwater recharge (Rau et al., 2020). This knowledge gap means that it is difficult to manage the resource sustainably.

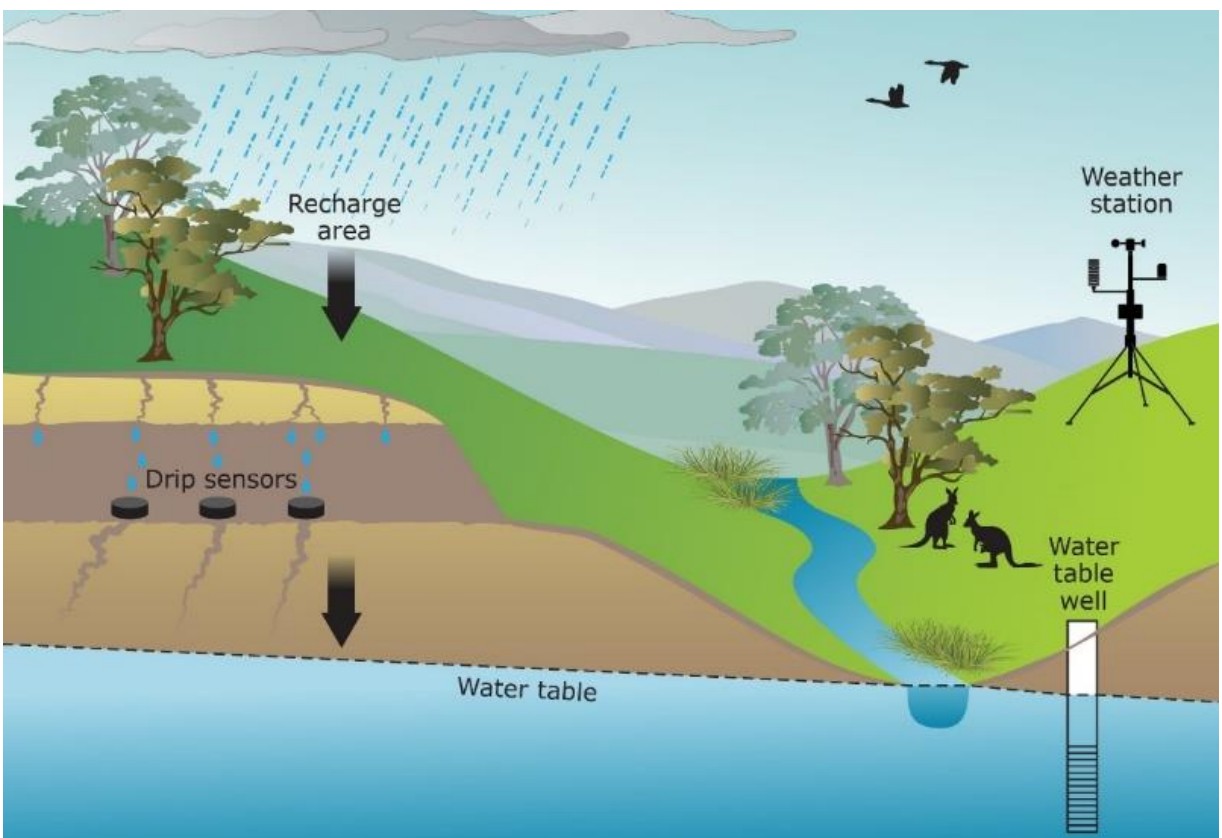

**Figure 1. Rainfall "events" drive water flow through the terrestrial environment, percolating through the ground - especially where the bedrock contains fractures that permit flow through the upper layers of soil to the water table below. The NGROS system has placed sensors in tunnels, mines and other subsurface spaces, represented here by the dark brown horizontal layer, to measure recharge over time and space and utilise locally available rainfall data to calculate "event-based" recharge following rainfall.**

Direct physical measurement identifying when groundwater recharge is occurring is possible by utilizing a sensor network of hydrological loggers deployed in underground spaces located between the soil and the aquifer (Fig. 1). Through measurements of water percolating into these spaces from above, we can record the groundwater recharge process in action. By using automated sensors, it is possible to determine when recharge events occur (which event, month, or season, and which climate condition). If rainfall event data is available, it is possible to
quantify the amount of rainfall needed to generate groundwater recharge. This is referred to as the 'rainfall

recharge threshold' and importantly, can vary temporally as well as spatially (e.g., Baker et al., 2021). Australia's National Groundwater Recharge Observing System (NGROS) provides the first dedicated sensor network for observing the recharge of groundwater at the event-scale across a wide range of geologies, environments, and climate types that represent a wide range of Australian hydroclimates. Utilizing tunnels, mines, and other subsurface spaces located between the soil and the groundwater table in the unsaturated zone, the sensors (Fig. 2) effectively record 'deep drainage', water that can move beyond the shallow subsurface and root zone to generate groundwater recharge. The NGROS has the temporal resolution to capture individual recharge events, with multiple sensors deployed at each site to constrain the heterogeneity of recharge between different flow paths, and to quantify (including uncertainty bounds) rainfall recharge thresholds. This infrastructure can identify that threshold wherever recharge is monitored. These data, openly accessible via a database, will inform the understanding of the recharge process, the most difficult term in the water balance (Ajami, 2021); improve water resource assessments, and improve the sustainable management of groundwater.

**2 The monitoring concept**

The monitoring concept for NGROS is to (1) provide data on the timing of recharge at the event timescale and (2) determine the amount of precipitation needed to generate recharge at an event scale. This concept focuses on the temporal aspects of groundwater recharge and does not directly provide estimates of recharge volume or amount. The observing system is distributed to cover a range of geologies, environmental and climate types, to quantify the relationship between these factors and the number of recharge events and the amount of precipitation needed to generate recharge. For logistical reasons, the NGROS was initially established in southern and eastern Australia, with the aim to expand the network over time. The sites focus on varied lithologies where water ingresses into the underground tunnel, mine or cave are observed. This is likely due to the presence of fractures, however the precise nature of the overlying fracture network is unknown. This is important for the success of the approach as it ensures sufficiently rapid, fracture-flow water movement that will preserve frequent event-based responses to recharge (as opposed to smoothed recharge through porous vadose zones). At locations within these sites where water movement was identified, sensors (drip counters) were deployed. At each site, between three and twelve sensors are deployed to provide a quantification of the heterogeneity of groundwater recharge. This number of sensors is guided by our previous experience monitoring in caves and karst where we have successfully interpreted recharge data using nine loggers in one cave system (Baker et al., 2021) and 14 loggers distributed across six caves in a regional analysis (Baker et al., 2020). While the regional groundwater at each site may also be influenced by regional groundwater flow, groundwater abstraction, and at some sites focused recharge from streams, the sensor network at each site strictly monitors the local vertical recharge through the vadose zone.

Where possible, NGROS site selection complements other initiatives that seek to understand vadose zone processes e.g., by co-location with Critical Zone Observatories or exiting groundwater monitoring bores or soil moisture monitoring infrastructure. The monitoring concept will benefit from combining the NGROS data with geophysical monitoring of the sites, as demonstrated by Campbell et al., 2017 and Leopold et al., 2020. Further examples of possible geophysical approaches, with an emphasis on karst geology, can be found in Ezersky et al., 2023. Furthermore, at selected sites, the NGROS project has also engaged with local stakeholders and communities to understand existing groundwater recharge knowledge. Our experience is that our monitoring

concept is technically straightforward and therefore suitable for citizen scientists and non-academics to maintain monitoring sites and undertake data collection with little training. In particular, this includes site staff at tourist mines and caves, and as demonstrated previously by Baker et al (2020), cavers and caving clubs. All groundwater recharge data will be made available through an open-access database and the findings from conversations with stakeholders will be shared through multiple means, including blog posts on the website and peer-review publications, as well as presentations with stakeholders.

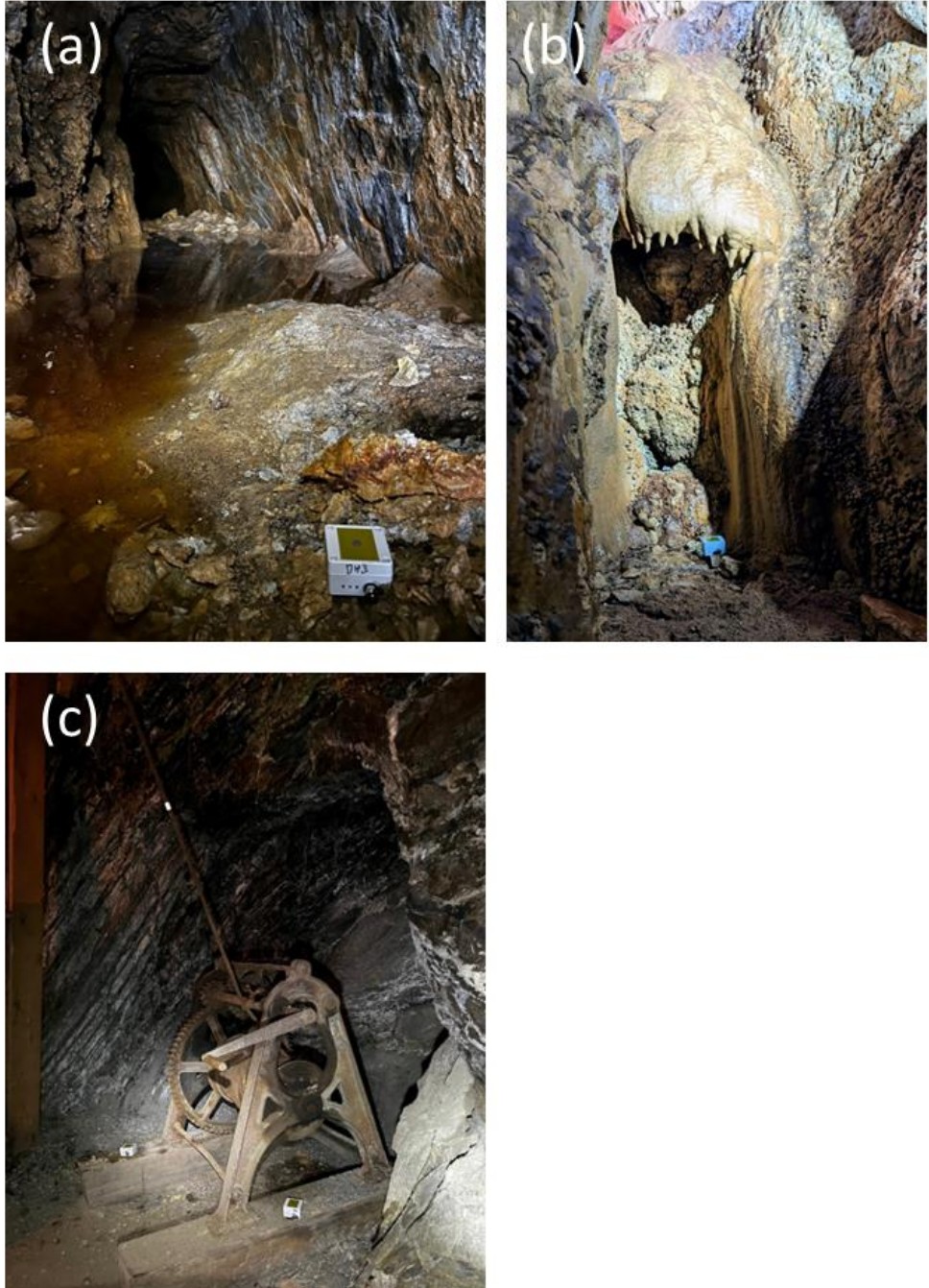

**Figure 2. Example logger deployment. (a) Durham Lead. The logger is situated on the floor, recording water dripping from the mine roof (b) Capricorn Caves. The logger is situated beneath a flowstone, an actively**

**forming calcium carbonate speleothem formed by flowing water. Water flows along the cave walls and drips onto the logger (c) Walhalla Long Tunnel. Two loggers are visible, both recording water dripping from the mine roof.**

### 2.1 Site descriptions

Fourteen NGROS sites were initially established in 2022, after consultation with potential partners that included National Parks, local and regional government, local and national speleological organizations, private landowners, and private operators of heritage and commercial mines, tunnels, and caves. Their location is shown in Fig. 3, and descriptions of these sites are provided below and summarized in Table 1. The NGROS project is continuing to expand on this initial network of sites. All sites were selected to ensure that the only source of recharge is from precipitation onto the land surface above the site.

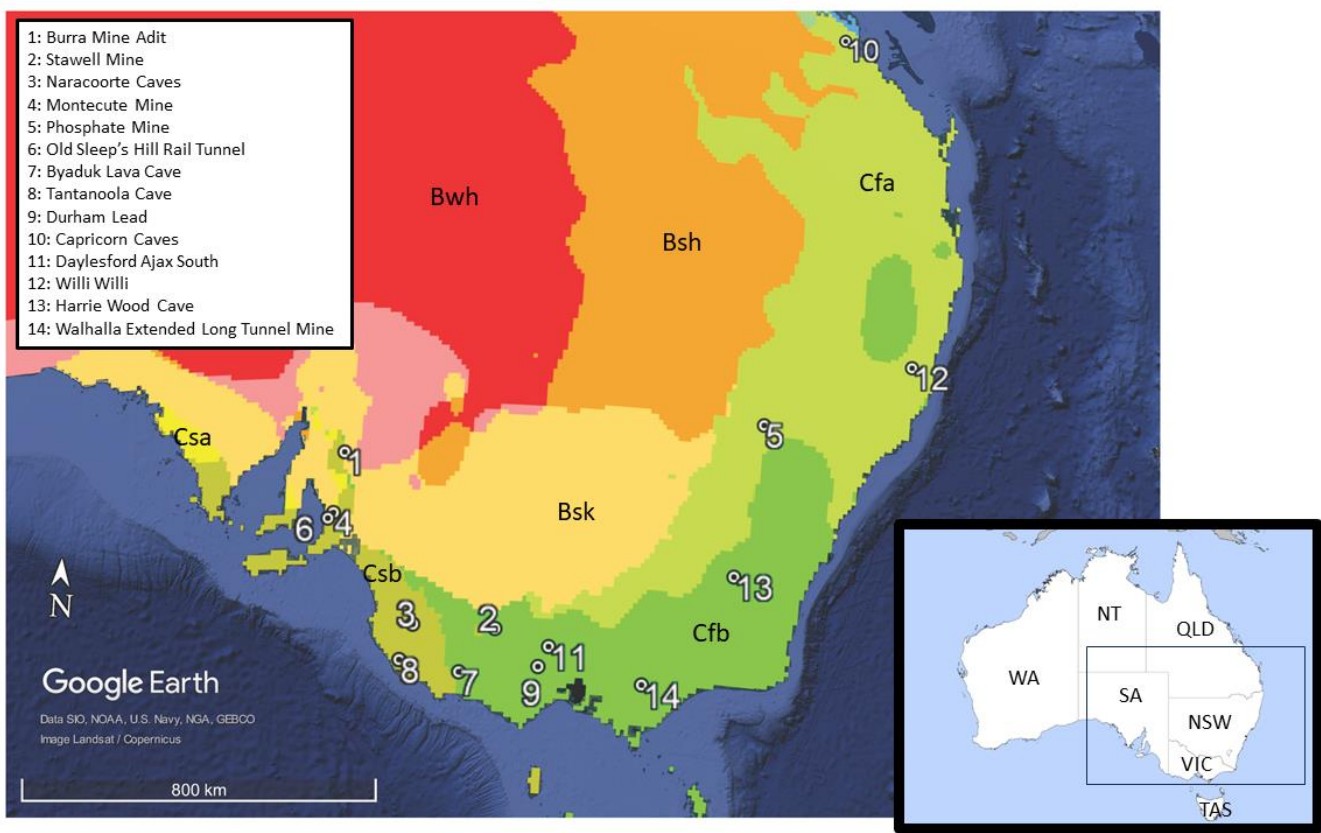

**Figure 3. Main image shows the location of the initial NGROS sites. Site numbers are in order of increasing precipitation and match the subsection headings for Sect. 2.2 The colours refer to Köppen–Gieger climate zones and from Peel et al (2007). The Google Earth basemap has image data provided by Landsat / Copernicus and map data from the SIO, NOAA, U.S. Navy, NGA and GEBCO, with the NGROS sites and the Köppen-Gieger classifications added as kml (the latter from Peel et al., 2007). Inset shows Australia, with the box defining the area of the main image. State boundaries are shown with the state and territory abbreviations. WA: Western Australia. SA: South Australia. NSW: New South Wales. VIC: Victoria. NT: Northern Territory. QLD: Queensland. The Australian Capital Territory (ACT) is not labelled. The outline map is from Wikimedia Commons, shared under the CC3.0 license.**

The sites are listed by increasing annual precipitation, with rainfall obtained from on-site Bureau of Meteorology weather stations with daily precipitation data where available, or from gridded daily precipitation using the Australian ~5 km by ~5 km resolution data (Jones et al., 2009) from the Australian Water Resources Assessment

Landscape model (AWRA–L) and accessed through the Australian Water Outlook website (awo.bom.gov.au). AWRA-L is a daily 0.05° gridded, distributed water balance model, which models water movement through the landscape from the initial precipitation through vegetation and soil moisture stores and losses of evapotranspiration, runoff and deep drainage. The current version (v7; Frost and Shokri, 2021; Frost et al., 2021) provides historical daily precipitation data (1985 CE onwards) that will be used by NGROS to determine rainfall recharge thresholds. The mean absolute error for daily precipitation in the gridded product is 1.2 mm for Australia as a whole, with highest uncertainties in the tropical north where rainfall extremes are highest and the monitoring network sparsest. This is outside of the initial NGROS monitoring region.

135

| Name | Location | Total annual P (mm) | Mean annual T (deg C) | Modelled annual ET (mm) | Site type | Geology age | Geology type | Vegetation | Soil characteristics | Köppen-Gieger climate classification |
|---|---|---|---|---|---|---|---|---|---|---|
| Burra Mine Adit, SA, Ngadjuri Country | 33.7° S, 138.9° E, ~500 m asl | 424 | 14.8 | 2211 | Disused copper Mine | Precambrian to Cambrian | Altered sedimentary | Inland Tall Shrublands | loamy | BSk (arid steppe, cold) |
| Stawell Mine, VIC, Djab Warrung Country | 37.6° S, 142.8° E, ~250 m asl | 480 | 14.2 | 1680 | Operating gold mine | Cambrian | Altered slate or schist | Dry scrub and woodland | Sodic clay (Sodosol) | BSk (arid steppe, cold) |
| Naracoorte Caves, SA, Meintangk Country | 37.0° S, 140.8° E, ~80 m asl | 485 | 14.3 | 1872 | Tourist caves | Miocene | Karstified limestone | Native scrub and woodland | sandy | CSb (temperate, dry summer, warm summer) |
| Montecute Mine, SA, Kaurna Country | 34.9° S, 138.7° E, ~150 m asl | 612 | 14.7 | 2090 | Disused metal mine | Precambrian to Cambrian | Sandstone | Woodlands with a Dense sclerophyll Shrub Understorey | Acidic loam over clay on rock | CSb (temperate, dry summer, warm summer) |
| Phosphate Mine, NSW, Wiradjuri Country | 32.6° S, 148.9° E, ~340 m asl | 620 | 17.2 | 2231 | Disused phosphate mine | Devonian | Karstified marmorised limestone | Native open woodland | Dermosol | Cfa (temperate, no dry season, hot summer) |
| Old Sleep's Hill Rail Tunnel, SA, Kaurna Country | 35.0° S, 138.6° E, 115 m asl | 700 | 16.8 | 2170 | Disused railway tunnel | Precambrian to Cambrian | sandstone | Native woodland | Loamy to clayey (Shallow soil on rock) | CSb (temperate, dry summer, warm summer) |
| Byaduk Lava Cave, VIC, Djab Warrung Country | 37.9° S, 142.0° S, ~150 m asl | 747 | 13.1 | 1600 | Lava cave | Quaternary | Basalt | Native scrub and open woodland | Organic loam soil (tenosol) | CSb (temperate, dry summer, warm summer) |
| Tantanoola Cave, SA, Bungandidj Country | 37.7° S, 140.5° E, ~50 m asl | 778 | 14.2 | 1692 | Tourist caves | Miocene | Karstified dolomite | Highly disturbed with remnant native grass, shrub and woodland | Sandy | CSb (temperate, dry summer, warm summer) |
| Durham Lead, VIC, Wadawurrung Country | 37.7° S, 143.9° E, ~360 m asl | 779 | 13.0 | 1678 | Disused gold mine | Ordovician | Metasediments | Native woodland | Sodic clayey (sodosol) | CSb (temperate, dry summer, warm summer) |
| Capricorn Caves, QLD, Darumbal Country | 23.16° S, 150.5° E, ~270 m asl | 815 | 23.4 | 2525 | Tourist caves | Devonian | Karstified reef limestone | Native dry rainforest | Bare rock | Cfa (temperate, no dry season, hot summer) |
| Daylesford Ajax South, VIC, Dja Dja Wurrung Country | 37.3° S, 144.1° E, ~550 m asl | 879 | 11.7 | 1728 | Disused gold mine | Ordovician | Metasediments | Grassland with scattered trees | Chromosol | Cfb (temperate, no dry season, warm summer |

| | | | | | | | | | | |
|---|---|---|---|---|---|---|---|---|---|---|
| Willi Willi, NSW, Djangadi Country | 30.9° S, 152.5° E, ~270 m asl | 1218 | 18.7 | 1879 | Cave | Permian | Karstified limestone | Remnant dry and subtropical rainforest | Dermosol | Cfa (temperate, no dry season, hot summer |
| Harrie Wood Cave, NSW, Ngarigo Country | 35.7° S, 148.5° E, ~980 m asl | 1260 | 11.1 | 1492 | Tourist cave | Silurian | Karstified limestone | Sub-alpine native woodland | Bare rock and thin rudosol | Cfb (temperate, no dry season, warm summer |
| Walhalla Extended Long Tunnel Mine, VIC, Gunaikurnai Country | 37.9° S 146.5° E, ~400 m asl | 1290 | 13.2 | 1565 | Tourist historic gold mine | Devonian | Metasediments | Native woodland | Thin chromosol | Cfb (temperate, no dry season, warm summer |

**Table 1. Descriptions of the fourteen initial NGROS sites. Sites are in order of increasing mean annual precipitation.**

Additional hydroclimate information for the sites is provided by historical modelled annual (synthetic) pan evaporation, obtained from the AWRA–L v7 output (1985-present). AWRA–L data is also used for mean annual temperature, except in the case that there is a Bureau of Meteorology weather station at the site. Climate classifications use the Köppen–Geiger system (Peel et al., 2007). Overall, the sites cover a mean annual temperature range from 11 to 24 °C with annual precipitation ranging from 424 to 1290 mm and modelled annual pan evaporation ranging from 1492 to 2525 mm. All sites are water-limited over annual timescales, with the modelled annual pan evaporation always greater than annual precipitation.

**2.1.1 Burra Mine Adit, SA, Ngadjuri Country (33.7° S, 138.9° E, ~500 m asl (above sea level))**

This short, hand-dug adit forms part of a copper mine complex which at its peak was the largest copper mine in Australia, and it is now part of a National Heritage Area. The mine lies within the Adelaide geosyncline, a thick Neoproterozoic to Cambrian sedimentary succession (Preiss, 2000). At the surface, there are loamy soils over rock (Hall et al., 2009), and a layer of fine loam was apparent at the edges of the adit entrance. The depth of the adit is approximately 5 m below ground where the shaft opens into the engine house (Wallis, L. pers. comm. 2022), dug through flaky sandstone. To take advantage of the deepest portion of the adit, five loggers were placed approximately 2 m apart along the sides of the adit beginning from the engine house end. Burra sits within a landscape of temperate grasslands. The local climate has a mean annual temperature of 14.8 °C, mean annual rainfall is 424 mm and modelled pan evaporation of 2211 mm. The climate according to the Köppen–Geiger system is BSk (arid steppe, cold). The site is currently under the management of the National Trust of South Australia.

**2.1.2 Stawell Mine, Stawell, VIC, Djab Warrung Country (37.6° S, 142.8° E, ~250 m asl)**

This mine adit is in an operating gold mine with large blasted and excavated access tunnels. The loggers are placed in an access tunnel at a level approximately 100 m below surface, suspended approximately 30 cm down from the fractured rock roof that is supported with rock-bolts, and some minor roof areas with reinforcing mesh. Sections

of the tunnel roof with shotcrete has been avoided. Gold-quartz veins here are associated with altered Cambrian slate or schist and metavolcanics adjacent to a Devonian intrusive body (Miller et al., 2006). The local climate has a mean annual temperature of 14.2 °C, mean annual rainfall 480 mm and modelled pan evaporation of 1680 mm. The climate is at the northern limit of Cfb (warm summer, no dry season, temperate), close to a transition to BSk (arid steppe, cold). The vegetation directly overlying the drip loggers is dry shrub and woodland with sodic clay subsoil (Sodosol) (Soil Health Knowledgebase, 2023). The mine site is privately owned, with Arete Capital Partners owning the majority share.

### 2.1.3 Naracoorte Caves, SA, Meintangk Country (37.0° S, 140.8° E, ~80 m asl)

Numerous shallow caves are present at Naracoorte, situated in the Miocene-age Mt Gambier limestone. The local climate has a mean annual temperature of 14.3°C, mean annual rainfall of 485 mm and modelled pan evaporation of 1872 mm. The climate is CSb (temperate, dry summer, warm summer). Loggers have been placed in two caves at locations where fracture-flow could be identified, with five loggers at irregular intervals in Blanche Cave and four loggers in Victorian Fossil Cave. Blanche Cave had a pine plantation above it for a period but is now revegetated with established native trees, and Victorian Fossil Cave is covered mostly by native scrub. This site is managed by National Parks and Wildlife Service of the Government of South Australia.

### 2.1.4 Montecute Mine, SA, Kaurna Country (34.9° S, 138.7° E, ~150 m asl)

This short, hand-dug adit is situated at the northern end of the metropolitan Adelaide region on private land and privately managed. It is unknown whether load-bearing ore was found in this adit, although cobalt, copper, gold, lead, nickel and silver are all found within a small radius of this site (Mining at Montacute, 2023). The adit is situated within a steep, wooded hillside dominated by native vegetation adjacent to a dolomite quarry and quartzite cliffs. Five loggers were placed at 1-2 m intervals beneath active drip sites starting approximately 10 m from the entrance to the adit. The local climate has a mean annual temperature of 14.7 °C, mean annual rainfall of 612 mm and modelled pan evaporation of 2090 mm. The climate is CSb (temperate, dry summer, warm summer).

### 2.1.5 Phosphate Mine, NSW, Wiradjuri Country ( -32.6° S, 148.9° E, 340 m asl).

With native box grassland vegetation cover, and mid-Devonian marmorized reef limestone geology, this site is located near the top of a shallow (~40 m above the nearby river alluvium) ridgeline formed by the reef limestone. The local climate has a mean annual temperature of 17.2 °C, mean annual rainfall 620 mm and synthetic pan evaporation of 2231 mm. The climate is Cfa (temperate, no dry season, hot summer). The only source of recharge is from precipitation onto the ridge above the site. This hand-dug heritage mine was developed for the possible extraction of mineralized bat-guano accumulated in karst passages. Loggers have been placed in the lowest mine passages, closest to the groundwater. This site complements existing long-term cave-based groundwater recharge monitoring at Cathedral Cave in the same formation (Jex et al., 2012), groundwater monitoring bores established at the site, a soil moisture network at the site (Berthelin et al., 2020), and the nearby (~10 km) UNSW Wellington Field Station (Rutlidge et al., 2023), and a newly established Critical Zone Observatory. This site is managed by the Government of New South Wales and Dubbo Regional Council.

### 2.1.6 Old Sleep's Hill Rail Tunnel, SA, Kaurna Country (35.0° S, 138.6° E, 115 m asl)

This disused, heritage-listed train tunnel is located on private land in the Adelaide metropolitan region. The tunnel was built in 1883 to carry commuters from the Adelaide central business district to suburbs in the inner Adelaide Hills region Southeast of the city centre, and for long-distance trains between Adelaide and Melbourne. The tunnel is constructed in the western foothills of the Mount Lofty Ranges with loamy to clayey soil over weathered sandstone (Hall et al., 2009). The tunnel has sandstone arches and piers lined with eight layers of brick, and was used for trains until 1919, after which it has been used for storage of valuable state possessions during World War II and commercial mushroom cultivation; it is now empty. The area above the tunnel is steep terrain with native soil and rock and largely consists of a conservation park for the remnant grey box eucalypt forest; loggers were placed about 10 m inside the tunnel at locations where drips were observed. The local climate has a mean annual temperature of 16.8 °C, mean annual rainfall of 700 mm and modelled pan evaporation of 2170 mm. The climate is CSb (temperate, dry summer, warm summer). This site is privately owned and managed.

### 2.1.7 Byaduk Lava Cave, VIC, Djab Warrung Country (37.9° S, 142.0° S, ~150 m asl)

This basalt lava cave is in Late Quaternary age basalts that derive from the volcanic eruption of Mt. Napier c. 36,000 years ago, the most recent active volcanism in Australia. The local climate has a mean annual temperature of 13.1 °C, mean annual rainfall of 747 mm and modelled pan evaporation of 1600 mm. The climate is CSb (temperate, dry summer, warm summer). The overlying vegetation is native shrub and open woodland with organic loam soil (Tenosol) (Soil Health Knowledgebase, 2023). The site is managed by the Government of Victoria through Parks Victoria and is located upstream of the Budj Bim Cultural Landscape, listed by UNESCO in 2019 (UNESCO World Heritage Convention, 2023)

### 2.1.8 Tantanoola Cave, SA, Bungandidj Country (37.7° S, 140.5° E, ~50 m asl)

This karst cave was formed in Miocene-age bryozoan dolomite, with sandy soils at the surface. The cave is set into a cliff face which is believed to have been exposed to wave action during a period of higher sea levels (DEH, 2008). Above the cave is a hilltop of native vegetation that has been highly disturbed by agricultural activities prior to the protection of the area as a conservation park in 1972. The local climate has a mean annual temperature of 14.2 °C, mean annual rainfall of 778 mm and modelled pan evaporation of 1692 mm. This site is managed by National Parks and Wildlife Service of the Government of South Australia.

### 2.1.9 Durham Lead, Clarendon Forest, VIC, Wadawurrung Country (37.7° S, 143.9° E, ~360 m asl)

This historic hand-dug adit located on private property is relatively short and of small diameter and is situated in gold deposits associated with Devonian quartz mineralisation of Ordovician sedimentary rocks. The local climate has a mean annual temperature of 13.0 °C, mean annual rainfall of 779 mm and modelled pan evaporation of 1678 mm. Native woodland and sodic clayey subsoil (Sodosol) overlie this adit. The climate is Cfb (temperate, no dry season, warm summer).

### 2.1.10 Capricorn Caves, QLD, Darumbal Country (23.16° S, 150.5° E, ∽ 270 m asl)

This natural cave site is in an isolated c. 50 m high limestone hill with 390M yr old Devonian reef limestone geology. It has a native dry rainforest vegetation cover. The local climate has a mean annual temperature of 23.4 °C, mean annual rainfall of 815 mm, and modelled pan evaporation of 2525 mm. The climate is on the boundary of Cfa (temperate, no dry season, hot summer) and tropical savanna climate classifications (Aw). Loggers have been placed in the lowest cave passages, closest to the groundwater. The site is privately owned and managed.

**2.1.11 Daylesford Ajax South, VIC, Dja Dja Wurrung Country (37.3° S, 144.1° E, ~550 m asl)**

This historic hand-dug adit, located on private property, is relatively short and of small diameter and associated with the gold deposits of the Castlemaine Supergroup (Lower Ordovician) sediments. The local climate has a mean annual temperature of 11.7 °C, mean annual rainfall of 879 mm and modelled pan evaporation of 1728 mm. Chromosol soils and scattered tall trees and grass overlies the adit. The climate is Cfb (temperate, no dry season, warm summer).

**2.1.12 Willi Willi, NSW, Djangadi Country (30.9° S, 152.5° E, 270 m asl).**

Daylight Cave is situated in a national park comprising cleared dry and subtropical rainforest with remnant vegetation. The site was burnt over by a containment (control) fire during the 2019/20 Australian wildfires. The limestone geology is Permian in age and comprises calcareous mudstone, crinoidal limestone and reef limestones. The ~60 km belt of limestone has been highly folded and faulted post-deposition, leading to the limestone retaining low primary porosity and becoming highly fractured. Adjacent to and overlying the cave is a soil moisture network comprising 20 randomly distributed soil moisture probes (an extension of the network described in Berthelin et al., 2020). Loggers are placed in the upper and lower levels of this cave to allow comparison between potential recharge events and soil moisture characteristics. The local climate has a mean annual temperature of 18.7 °C, mean annual rainfall of 1218 mm and modelled pan evaporation of 1879 mm. The climate is Cfa (temperate, no dry season, hot summer). The site is managed by the Government of New South Wales through the NSW National Parks and Wildlife Service.

**2.1.13 Harrie Wood Cave, NSW, Ngarigo Country (35.7° S, 148.5° E, ~980 m asl)**

This former tourist cave is situated in Silurian limestone that has low primary porosity and is highly fractured and is a long-term hydrological monitoring site (Markowska et al., 2015; Chapman et al 2023). Located in the Kosciuszko National Park, the local vegetation comprises sub-alpine open snow gum woodland, although above the cave the steep slopes limit the extent of soil cover and vegetation. The site was burnt over in the 2019/20 Australian wildfires. This site provides data from a sub-alpine location that experiences seasonal snowfall, a site that is recovering from wildfire, and a site from which stalagmite paleoclimate records have been obtained (Tadros et al., 2022). Loggers are located on a depth transect from the cave entrance to a depth approximately 50 m below land surface. The local climate has a mean annual temperature of 11.1 °C (recorded at the on-site weather station), mean annual rainfall of 1260 mm and modelled pan evaporation of 1492 mm. The climate is Cfb (temperate, no dry season, warm summer). The site is managed by the Government of New South Wales.

**2.1.14 Walhalla Extended Long Tunnel Mine, VIC, Gunaikurnai Country (37.9° S 146.5° E, ~400 m asl)**

Walhalla is a large-scale historic goldmine, with one level managed as a tourist site:  lower levels of the historic mine are flooded with groundwater. The gold deposits are associated with Devonian age dykes found within the Devonian metasedimentary rocks of the eastern Victorian gold province. The local climate has a mean annual temperature of 13.2 °C, mean annual rainfall of 1290 mm and modelled pan evaporation of 1565 mm. Native forest with tall trees and thick understory occurs on the slopes with thin soils (Chromosol) above the mine. The climate is Cfb (temperate, no dry season, warm summer). The site is a crown land reserve and is managed by the Victoria Government through the Walhalla Board of Management.

**2.2. Technical description of loggers**

The NGROS uses commercially available Stalagmate © loggers as shown in Fig. 2. These use a pressure transducer within a sealed unit to detect water dripping onto their sealed surface, with hardware detecting drip impacts over noise (e.g., vibrations). The units use a TinyTag© body manufactured by Gemini Data Loggers UK Ltd. The loggers count drips over a user specified period that could range from seconds to days. Based on the observed frequency of drips and the remoteness and ease of access of the sites, we have chosen to count the total number of drips over time periods ranging between 0.25 and 1.0 hours. This provides sufficient temporal resolution to identify recharge events and at least 11 months of data collection until memory capacity is reached and a download is required. Memory capacity is 4 years for data collected at hourly intervals, high energy 3.6V half-AA cells provide power for at least 3 years of continuous deployment. Data output is in the form of .csv or .xls files and data processing only involves data screening in the case that the logger is moved. Detailed analysis of logger precision is provided in Collister and Mattey (2009). Manufactured initially for cave scientists recording water flux to stalagmites (Collister and Mattey, 2009), they have previously been demonstrated to be suitable for long-term deployment in underground spaces. For example, loggers deployed to investigate the heterogeneity of recharge by Jex et al., (2012) are still in service despite the cave chamber being flooded by groundwater multiple times, as are those used in the hydrological characterisation of flow paths (Markowska et al., 2015), to quantify the effects of wildfire on recharge event hydrographs (Bian et al., 2019) and to quantify groundwater recharge thresholds and soil and unsaturated zone storage (Baker et al., 2020, 2021). Published applications of the use of the loggers for their original application of monitoring individual cave drip waters are global in extent e.g., in China (Hu et al., 2008), USA (Onac et al., 2008; Wortham et al., 2021) and Europe (Surić et al., 2017).

**3    Initial results from the first year of monitoring**

As examples of the types of data and interpretations that are possible, we present one-month of data from three contrasting sites from the first year of monitoring. At the Byaduk Lava Cave (VIC), Fig. 4 shows five recharge events that occurred in October 2022. This included four events that occurred over a 10 day period. At this site, recharge occurs within 24 h of rainfall. This allows for event-scale determination of the amount of rainfall necessary to generate recharge. Because rainfall data is at daily resolution and individual rain events can occur over two recording days, we calculate the 48 h total rainfall recharge thresholds. At the site, in Austral spring, these thresholds vary from 32.2 mm prior to the first recharge event to 11.0 mm prior to the fifth event. A decreasing amount of rainfall needed to generate recharge over time is observed over the four recharge events that occur over 10 days.

At the Phosphate Mine (NSW), Fig. 5 shows three recharge events that occurred in the month of April 2022. At this site, recharge also occurs within 24 hours of rainfall, although not all monitoring locations record recharge at all events. The 48 h total antecedent rainfall varies between 36.2 and 19.4 mm. During April 2022 there are 48 h rainfall amounts of ~10 to 17 mm and these do not generate recharge.

At Naracoorte Caves (SA), Fig. 6 presents two recharge events in October 2022. At this site, recharge again occurs within 24 hours of rainfall occurring. In contrast to the other sites, a complex recharge response to rainfall can be observed. Although the two monitoring sites increase in water flux, there is a non-linear response to the second recharge event, with some inverse correlations in flow rates occurring at the two monitoring sites that are indicative of flow-switching. For these two recharge events, the 48 h rainfall recharge thresholds vary between 31.2 and 15.0 mm. However, a subsequent rainfall event with ~16 mm of rainfall did not generate further recharge at these monitoring sites.

Our example NGROS data demonstrates how rainfall recharge thresholds can be obtained at an event timescale. As data accumulates over time, we anticipate being able to investigate how rainfall recharge thresholds vary with season and under different climate drivers. With this example, with data analysed from the Austral spring and autumn, we observe a similar range in 48 h antecedent rainfall that is required to generate recharge at sites with contrasting geology (basalt, fractured marble, and porous limestone) and water-limited sites that have total annual precipitation < 750 mm and modelled pan evaporation at least twice that amount. At Byaduk (Fig. 4), we believe this is the first data presented using drip loggers in the unsaturated zone of basalt.

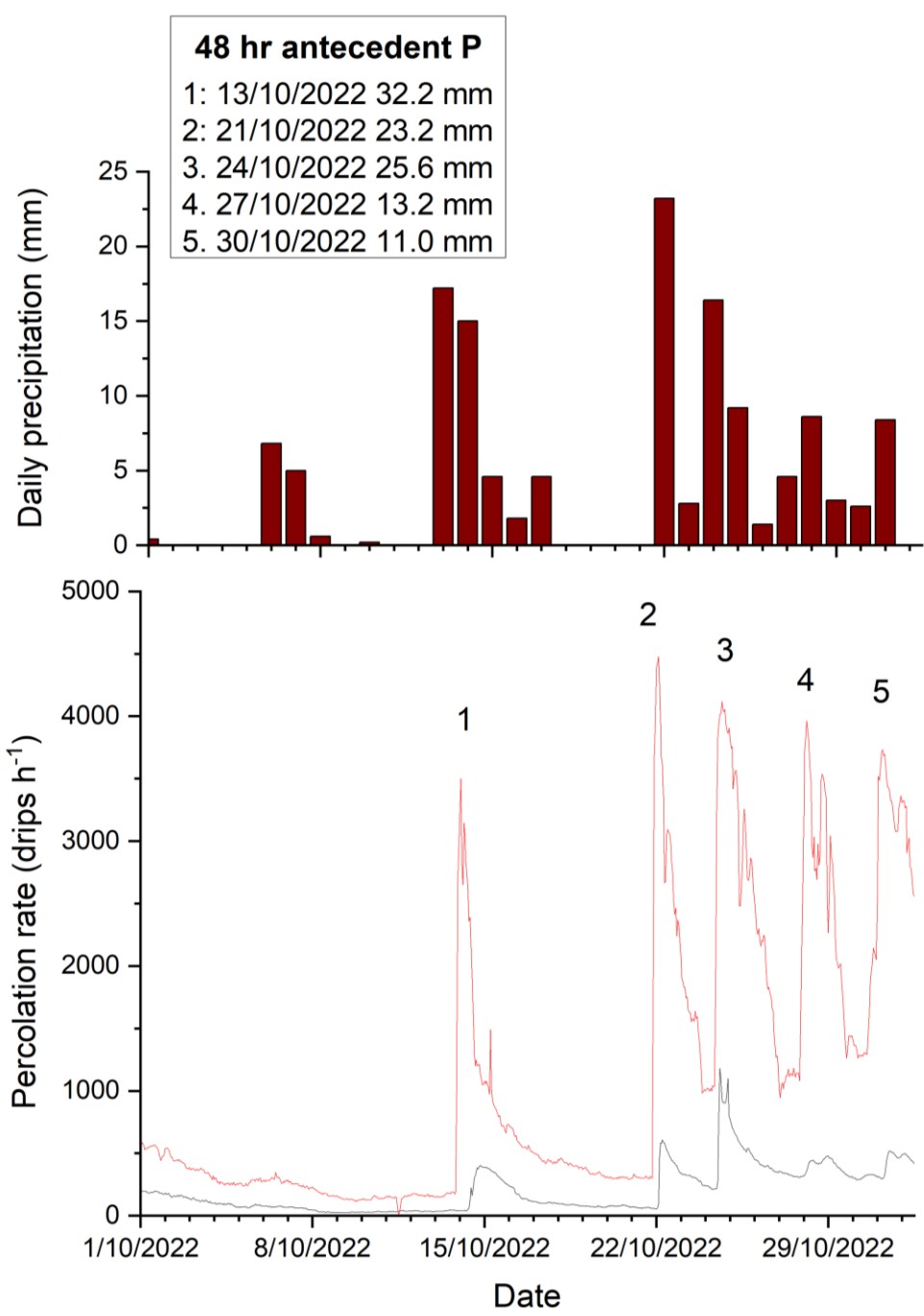

**Figure 4. Byaduk Lava Cave. Top: daily precipitation. Bottom: percolation rate recorded by two loggers in the lava cave (shown by red and black lines). Five recharge events are recorded, the 48-hour antecedent precipitation shown in the inset table.**

330

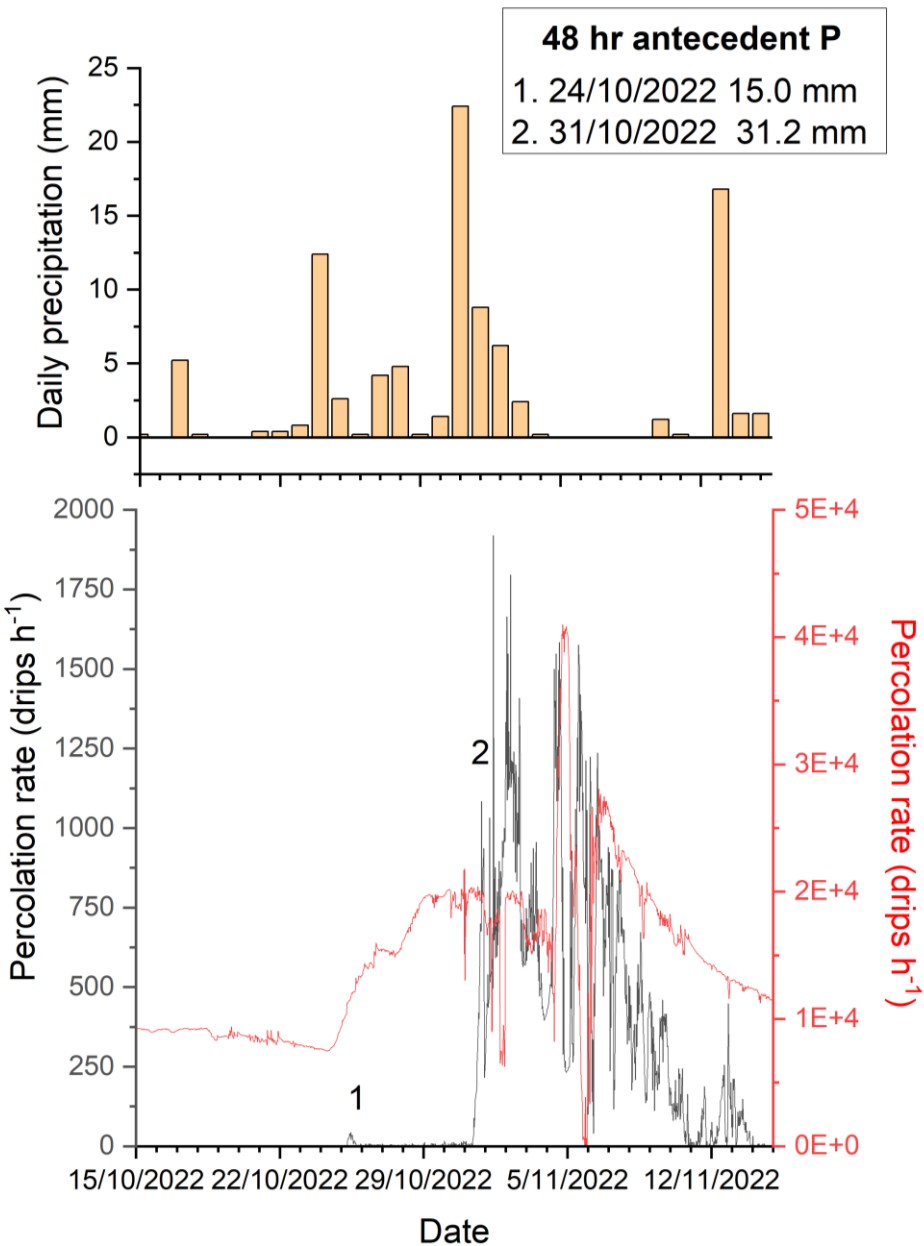

335 **Figure 5. Naracoorte Caves. Top: daily precipitation. Bottom: percolation rate recorded by two loggers in the cave (shown by red and black lines). Note the different y-axis scales. Two recharge events are recorded, the 48-hour antecedent precipitation shown in the inset table.**

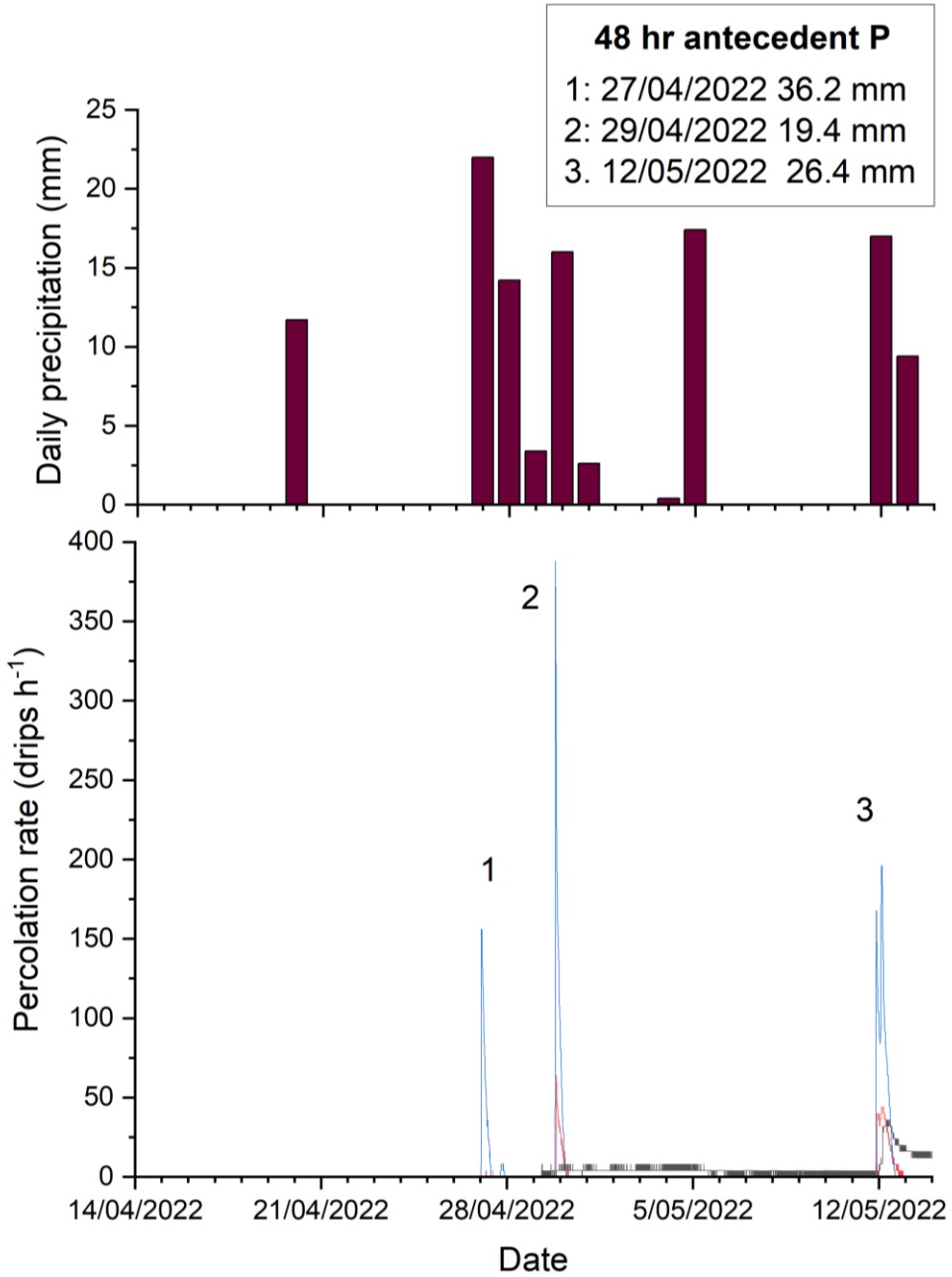

**Figure 6. Phosphate Mine. Top: daily precipitation. Bottom: percolation rate recorded by three loggers in the cave (shown by red, blue and black lines). Three recharge events are recorded, the 48-hour antecedent precipitation shown in the inset table.**

**4 Summary and Outlook**

The NGROS is established in Australia, a country where groundwater is of particular importance, and the concept could be replicated elsewhere in the world and that it could benefit any country where a greater understanding of the timing of groundwater recharge is of importance. In Australia, in 2013 it was estimated that groundwater was worth AU$ 6.8 billion Gross Domestic Product equivalent to the Australian economy (National Centre for Groundwater Research and Training, 2013), mainly supporting extractive minerals and agricultural activities. Currently, groundwater makes up around 17% of accessible water in Australia and accounts for more than 30% of total water consumption. This is expected to increase as surface water resources diminish due to climate change

and prolonged drought (Enemark et al., 2019). Indeed, the recent 'Millennium' and 2016-2019 droughts have shown how groundwater provides essential water security in water-scarce regions of Australia, with the use of groundwater increasing drastically over the course of the Millennium drought, accounting for as much as 70% of irrigation water across the Murray Darling Basin (Richardson et al., 2011).

Due to Australia's dependence on finite groundwater resources and the mounting pressure on it from a warmer, and in some places drier, climate, as well as expanding agricultural and mining developments, groundwater and drought resilience is a particular focus of the Commonwealth Scientific and Industrial Research Organisation (CSIRO), Australia's national science research agency. Real-time monitoring of the timing of groundwater recharge variability provided by NGROS will help further the science around hydroclimate variability, drought

frequency and intensity, as well as other climate extremes to complement and support groundwater research being undertaken at CSIRO. Particularly their Drought Resilience Mission, in which the goal is to reduce the impacts of drought in Australia by 30% by 2030 (https://www.csiro.au/en/about/challenges-missions/Drought-Resilience). This data generated is predominantly of the timing of recharge at the event timescale and the determination of rainfall recharge thresholds. The novelty of the NGROS approach is that it provides data on the timing of recharge

at the event timescale at sites where the source of the recharge water is known e.g. from direct and focused recharge from precipitation. It therefore permits the determination of the amount of precipitation needed (i.e. the rainfall threshold) to generate recharge at an event scale. As demonstrated in Figure 4, the observing system has demonstrated the capability to determine rainfall recharge thresholds at an event scale from a range of geologies, environments, and climate zones. The heterogeneity of recharge can also be investigated. The observing system

precludes the direct determination of recharge volume or amount of recharge, which would require further analyses. However, data generated on the timing of recharge at the event timescale can be combined with other datasets that estimate recharge volume such as the soil moisture water balance, chloride mass balance or isotopic techniques (Jasechko, 2019; Crosbie and Rachakonda, 2021; Berthelin et al., 2023). The NGROS data will complement the data collected by other field infrastructure estimating vadose zone water fluxes such as the newly

funded Australian Critical Zone Observatory network (https://www.tern.org.au/critical-zone/) and field point verification for nationally gridded hydrological data products such as the Australian Water Resources Assessment Landscape model (AWRA–L).

Our vision with this project in the long term is to contribute to the conversations about groundwater recharge, sustainable groundwater management and to understand what stakeholders know about groundwater. We plan to

increase the spatial scale of the network by adding more sites nationally, and potentially form part of an international network of similar infrastructure. We also aim to maintain the network for as long as possible, to enable the collection of recharge data relevant to the predicted rainfall intensification with climate change (Fowler et al., 2021) and under different climate drivers (e.g., changes in the southern Annular Mode and El Niño – Southern Oscillation). With multi-year data, we will be able to quantify the annual total recharge at each of our

monitoring sites and investigate the relationship between that and annual climate parameters. This approach will also allow the investigation of recharge characteristics at sites where fracture flow is not dominant and individual event-based recharge characterization is not possible. Finally, by maintaining a high temporal resolution of sampling (hourly or more frequent), the data generated by NGROS can be used to identify recharge pathways e.g., time-series analysis to identify non-linear flow behaviour typical of karst systems (Baker and Brunsdon, 2003)

and identify the role of tree water use where daily drip rate fluctuations occur at times of tree water stress (Coleborn et al., 2016).

The data generated by the network will support training opportunities for students, researchers, and stakeholders, providing data about groundwater recharge across Australia that is essential to understand what is possible in terms of management. The engagement with stakeholders that depend on groundwater, but also with managers

and people that interact with spaces of groundwater (e.g., caves), will give us the opportunity to map out knowledges and possibilities to build capacity for maintaining monitoring systems. A critical consideration relating to the data collected by NGROS is its translation in terms of informing human-groundwater relations. In particular, how does the NGROS contribute to broader groundwater literacy in Australia? As Baldwin et al., (2012) have noted, for groundwater planning to be effective, there is a critical need for scientists to *exchange* knowledge

and understanding with local communities (Baldwin et al., 2012). This has not always happened. As they further note there is a need for all relevant knowledge to be understood and considered effectively by key stakeholders in "an open, inclusive and transparent decision-making process" as well as "integrating local and Indigenous knowledge" into groundwater considerations and assessments" (Baldwin et al 2012, 75). In an Australian context – but also many contexts globally – First Nations knowledges and relationships with groundwater is a key

consideration. Moggridge (2020, 23) in his research on Australian groundwater, for example, has observed how "Over many thousands of years, Aboriginal people have accumulated a comprehensive and astounding knowledge of groundwater" that needs to be taken in consideration by governments and groundwater managers. These groundwater knowledges have often been learnt and shared in Dreamtime stories relating to sites being created and then passed on from one generation to another (Moggridge, 2020). An area of interest for the NGROS project

is to learn from ancient knowledges of groundwater and use tools, like the geoscientific instrumentation used for this project, to better interact and manage groundwater. As the current sample of sites shows, in addition to each site having its own unique physical characteristics, they are on Indigenous Countries under various ownership and management regimes (e.g., government, corporate, family). Understanding these complex layers of history, management and the physical environment is the intention of the NGROS.


*Data availability.* The preliminary data shown in this article is available within the larger NGROS database accessible from https://groundwater.unsw.edu.au.

*Author Contributions.* The observing system and associated database were designed and implemented by AB, MS, WT, MA and MMZ. Funding was obtained by AB, MS, WT, MA and MMZ. The paper was written and conceived

by AB, MS, WT, MA, MMZ, and SP.

*Competing Interests.* The authors declare that they have no conflict of interest.

*Acknowledgements.* Intersect Australia Ltd are acknowledged for database implementation. We thank the organizations supporting our research sites, including Capricorn Caves, Dubbo Regional Council, the Kempsey Speleological Society, the National Parks and Wildlife Service in New South Wales and South Australia, Parks

Victoria, Mine Management at Stawell and Walhalla, and Red Rock Australasia. Andreas Hartmann and Romane Berthelin provided soil moisture monitoring equipment to establish the Macleay site. Sirjana Adhikari, Vivian Zhang and Uli Bauer assisted at Victorian research sites. Anna Blacka drew Figure 1.

*Financial Support.* The Australian Research Council Large Infrastructure, Equipment and Facilities grant scheme (LE220100125) funded this research, with funding contributions from Flinders University, Deakin University and

UNSW. MMZ is additionally supported by a UNSW research grant.

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
