# Peer review of "An underground drip water monitoring network to characterize rainfall recharge of groundwater at different geologies, environments, and climates across Australia"

_EGUsphere, 2023_

## Author Response (AR1)

We would like to thank the reviewers for their comments on our manuscript 'An underground drip water monitoring network to characterize rainfall recharge of groundwater at different geologies, environments, and climates across Australia'. Please find our point-by-point response below, with the reviewer comment in italics and our revised text in red.

**In response to RC1**:

*General comment: 'The lack of a quantitative derivation based on the drip counting thus limited the data processing to the frequency domain.'* We agree. As we wrote in AC2, and as stated on lines 74-75 of the pre-print, the monitoring concept for NGROS is to (1) provide data on the timing of recharge at the event timescale and (2) determine the amount of precipitation needed to generate recharge at an event scale. We agree with the reviewer that this concept focuses on the temporal aspects of groundwater recharge. We agree that the focus is on the frequency domain, and the approach precludes the determination of recharge volume or amount of recharge, which would require further analyses that are outside the scope of this pre-print. We provide additional clarification about the recharge data that will be provided by the NGROS on line 75-76:

This concept focuses on the temporal aspects of groundwater recharge and does not directly provide estimates of recharge volume or amount.

And this is reemphasized on line 370:

Real-time monitoring of the timing of groundwater recharge variability provided by NGROS will help further the science around hydroclimate variability, drought frequency and intensity...

And see also our response to CC1 and our addition of further novelty statements about this approach from line 381 (using references that are already cited in the pre-print):

The observing system precludes the direct determination of recharge volume or amount of recharge, which would require further analyses. However, data generated on the timing of recharge at the event timescale and the determination of rainfall recharge thresholds can be combined with other datasets that estimate recharge volume such as the soil moisture water balance, chloride mass balance or isotopic techniques (Jasechko, 2019; Crosbie and Rachakonda, 2021; Berthelin et al., 2023).

*General comment: 'Another drawback related to the presentation of the NGROS system is that the dataset recorded at the observation sites is not available despite the authors claimed its availability from a specified website'.* We apologize for this. We timed the submission of the preprint for the expected launch date of the database, but unfortunately mis-timed this by two weeks due to unexpected delays while unanticipated cybersecurity checks were made. We confirm that the database has been publicly available since mid-October, unfortunately this was a few days after this review was made. We have updated the url (line 427):

The preliminary data shown in this article is available within the larger NGROS database accessible from https://groundwater.unsw.edu.au.

*Specific comment: The sub-sections in section 2.1 are hard to read given the large number of monitoring sites (14!). A summarizing table with all climatic, geological and landcover features far all sites will be useful for a better clarity.* We agree and we have added a summary table (Table 1) at line 140, which we refer to on line 109. This was also requested by the second reviewer RC2. For ease of reading, Table 1 is appended at the end of this response letter.

*Specific comment: Moreover, it seems that the comments regarding Figure 5 and 6 are not complete, compared to those regarding Figure 4 (lines 306-312).* We did not find this to be the case and could not find anything to correct here. As we wrote in AC2 "We checked whether the text relating to Figures 4, 5 and 6 was complete, and at this time would not propose to add any further text about these three examples."

*Specific comment: The results section should deal with all monitoring sites, particularly given the unavailability of observation records.* The data is available (see our response above). As we wrote in our public reply (AC2), "we appreciate the request to provide results from all 14 initial NGROS sites. However, we believe that level of data analysis and interpretation would be outside the scope .... We would like to note that the manuscript focuses on the initial fourteen sites and is intended to present the monitoring concept and research platform that is NGROS, including three examples illustrating the type of data collected. We expect to expand the number of sites over time as more partners join to collaborate with the project."

**In response to RC2**:

*Comment: Figure 1: describe the dark brown layer in which the drip sensor are located.* We edit the caption to Figure 1 as requested:

**Figure 1. Rainfall "events" drive water flow through the terrestrial environment, percolating through the ground - especially where the bedrock contains fractures that permit flow through the upper layers of soil to the water table below. The NGROS system has placed sensors in tunnels, mines and other subsurface spaces, represented here by the dark brown horizontal layer, to measure recharge over time and space and utilise locally available rainfall data to calculate "event-based" recharge following rainfall.**

*Comment: L79: "The sites focus on fractured-rock lithologies;". That means the drip sensors should be located at the intersects with the major fractures that transport the water downwards. How can this be ensured/tested?* We clarify the relationship between the logger locations and overlying fractures on lines 80-82:

The sites focus on varied lithologies where water ingresses into the underground tunnel, mine or cave are observed. This is likely due to the presence of fractures, however the precise nature of the overlying fracture network is unknown. This is important for the success of the approach as it ensures sufficiently rapid, fracture-flow water movement that will preserve frequent event-based responses to recharge (as opposed to smoothed recharge through porous vadose zones).

*Comment: L83-84: "At each site, between three and twelve sensors are deployed …"* How to make sure the number is *large enough to provide representative information?* We provide the context for the number of loggers at each site on lines 87-89, using two papers that are already cited:

This number of sensors is guided by our previous experience monitoring in caves and karst where we have successfully interpreted recharge data using nine loggers in one cave system (Baker et al., 2021) and 14 loggers distributed across six caves in a regional analysis (Baker et al., 2020).

*Comment: Figure 3: the caption is really long. Could you at least provide a legend and explain the climate zones directly in the map?* We have added a legend to Figure 3 to show the sample site numbers, and have added the climate zones to the map. This enables the shortening of the figure caption as requested. The revised Figure 3 and caption are:

[Figure]

**Figure 3. Main image shows the location of the initial NGROS sites. Site numbers are in order of increasing precipitation and match the subsection headings for Sect. 2.2.**  **The colours refer to Köppen–Gieger climate zones and from Peel et al (2007).**  **The Google Earth basemap has image data provided by Landsat / Copernicus and map data from the SIO, NOAA, U.S. Navy, NGA and GEBCO, with the NGROS sites and the Köppen-Gieger classifications added as kml (the latter from Peel et al., 2007). Inset shows Australia, with the box defining the area of the main image. State boundaries are shown with the state and territory abbreviations. WA: Western Australia. SA: South Australia. NSW: New South Wales. VIC: Victoria. NT: Northern**

Territory. QLD: Queensland. The Australian Capital Territory (ACT) is not labelled. The outline map is from Wikimedia Commons, shared under the CC3.0 license.

*Comment: L121 onwards: is there any information on the precision of the AWRA-L precipitation data (some RMSE or std)? Alternatively, are there some local stations at some of the sites to give an idea about the strengths/weaknesses of the product?* We provide additional information on the precision of the gridded precipitation product available in Australia on line 135-137:

The mean absolute error for daily precipitation in the gridded product is 1.2 mm for Australia as a whole, with highest uncertainties in the tropical north where rainfall extremes are highest and the monitoring network sparsest. This is outside of the initial NGROS monitoring region.

*Comment: L137/2.1.1 to 2.1.14: The site summaries are written systematically and short. It's still a bit tricky to understand which of them are similar or where they differ. Could you come up with a summary table?* We add a summary table (Table 1) as also requested by RC1. This is appended at the end of this response letter.

*Comment: In Fig 6, it seems that the drip sensor resolution is coarser than 1 drip/hour. Can you comment on the precision of these devices and or the data processing (would suit into 2.2).* We provide more technical details about the loggers as requested on lines 292-299:

The units use a TinyTag© body manufactured by Gemini Data Loggers UK Ltd. The loggers count drips over a user specified period that could range from seconds to days. Based on the observed frequency of drips and the remoteness and ease of access of the sites, we have chosen to count the total number of drips over time periods ranging between 0.25 and 1.0 hours. This provides sufficient temporal resolution to identify recharge events and at least 11 months of data collection until memory capacity is reached and a download is required. Memory capacity is 4 years for data collected at hourly intervals, high energy 3.6V half-AA cells provide power for at least 3 years of continuous deployment. Data output is in the form of .csv or .xls files and data processing only involves data screening in the case that the logger is moved. Detailed analysis of logger precision is provided in Collister and Mattey (2009). Manufactured initially for cave scientists recording water flux to stalagmites (Collister and Mattey, 2009), they have previously been demonstrated to be suitable for long-term deployment in underground spaces.

*Comment: Chapter 4 (Summary and Conclusions): The network is located within Australia so it makes sense highlighting its importance for Australian water research and education. However, similar networks could also be of great benefit for other regions where groundwater (and its recharge dynamics) are important. Could you add some words about why such networks would benefit other regions, too?* We add some additional text on lines 356-358 to confirm that the NGROS could be adapted elsewhere in the world. Also, further benefit and novelty statements are included in response to CC1 in the same section (see below). On lines 356-358:

The NGROS is established in Australia, a country where groundwater is of particular importance, and the concept could be replicated elsewhere in the world and that it could benefit any country where a greater understanding of the timing of groundwater recharge is of importance.

**In response to CC1:**

*Comment: It would be good if the authors can further clarify whether or not this approach can give quantitative estimates of recharge rates or volumes (as flagged by the referee), and if the maximum value of the data lies in combining the drips with other data from instrumentation at CZO sites.* We provide additional text to confirm that the novelty of the approach focuses on temporal aspects of recharge and that this can be combined with methods that estimate recharge volume. Lines 381-385 now read:

The observing system precludes the direct determination of recharge volume or amount of recharge, which would require further analyses. However, data generated on the timing of recharge at the event timescale and the determination of rainfall recharge thresholds can be combined with other datasets that estimate recharge volume such as the soil moisture water balance, chloride mass balance or isotopic techniques (Jasechko, 2019; Crosbie and Rachakonda, 2021; Berthelin et al., 2023).

*Comment: More discussion in this section of how this application of drip loggers extends their use from what has been done before, and how this new data source will add to our current literature-based understanding would be welcome.* We provide more information on the novelty and wider application of our approach on lines 375-379:

This data generated is predominantly of the timing of recharge at the event timescale. The novelty of the NGROS approach is that it provides data on the timing of recharge at the event timescale at sites where the source of the recharge water is known e.g. from direct and focused recharge from precipitation. It therefore permits the determination of the amount of precipitation needed (i.e. the rainfall threshold) to generate recharge at an event scale.

**Additional revision**

As stated in our response to RC1, a copy and paste error meant that incorrect text was originally provided for Capricorn cave. We have replaced the incorrect site description text for Capricorn Cave. The correct text is on lines 235-240.

[revised manuscript text omitted]

---

## Author Response (AR2)

We thank the Associate Editor for their review and comments on our revised manuscript. We have found them helpful and have implemented the two suggestions.

We confirm that our approach is suitable for contributions by non-professionals, as we have had first-hand experience of that. We add new text on lines 98-102:

Our experience is that our monitoring concept is technically straightforward and therefore suitable for citizen scientists and non-academics to maintain monitoring sites and undertake data collection with little training. In particular, this includes site staff at tourist mines and caves, and as demonstrated previously by Baker et al (2020), cavers and caving clubs.

We have also strengthened the text to stress the importance of a combined geophysical component. On lines 93-97 we include the reference suggested by the Asociate Editor together with two specific examples of where geophysical methods have been used combined with percolation water hydrology.

The monitoring concept will benefit from combining the NGROS data with geophysical monitoring of the sites, as demonstrated by Campbell et al., 2017 and Leopold et al., 2020. Further examples of possible geophysical approaches, with an emphasis on karst geology, can be found in Ezersky et al., 2023.

We add the two new references to the reference list on lines 454 and 476.

Finally, on line 457 we take the opportunity to update one reference which is now published.